# Teaching Models to Understand (but not Generate) High-risk Data

**Ryan Wang**[1]    **Matthew Finlayson**[1]    **Luca Soldaini**[2]
**Swabha Swayamdipta**[1]    **Robin Jia**[1]

[1]Department of Computer Science, University of Southern California;
[2]Allen Institute for AI

{ryanywan,mfinlays}@usc.edu

## Abstract

Language model developers typically filter out high-risk content—such as toxic or copyrighted text—from their pre-training data to prevent models from generating similar outputs. However, removing such data altogether limits models' ability to recognize and appropriately respond to harmful or sensitive content. In this paper, we introduce Selective Loss to Understand but Not Generate (SLUNG), a pre-training paradigm through which models learn to understand high-risk data without learning to generate it. Instead of uniformly applying the next-token prediction loss, SLUNG selectively avoids incentivizing the generation of high-risk tokens while ensuring they remain within the model's context window. As the model learns to predict low-risk tokens that follow high-risk ones, it is forced to understand the high-risk content. Through our experiments, we show that SLUNG consistently improves models' understanding of high-risk data (e.g., ability to recognize toxic content) without increasing its generation (e.g., toxicity of model responses). Overall, our SLUNG paradigm enables models to benefit from high-risk text that would otherwise be filtered out.

## 1   Introduction

Pre-training data plays a crucial role in shaping the capabilities and behavior of language models. Current data curation practices focus on creating clean, high-quality training corpora that reflect the kind of text models are expected to generate. This has led to extensive efforts to filter high-risk content—such as harmful, misleading, or copyrighted material—from pre-training datasets, so that models will not learn to generate them (Grattafiori et al., 2024; Soldaini et al., 2024; Liu et al., 2024).

However, discarding risky data altogether reduces the models' ability to recognize and respond to risky inputs, leaving them ill-equipped for real-world deployment. For example, pre-training on toxic data improves a language model's ability to effectively recognize and handle harmful requests in practice—an ability that cannot be developed from training on clean data alone (Longpre et al., 2023; Rae et al., 2022; Welbl et al., 2021).

In this work, our goal is to obtain the best of both worlds: we train models that can understand high-risk[1] data without being able to generate it. To achieve this, we move beyond the standard next-token prediction objective, which inherently entangles the model's ability to generate and its ability to understand. We introduce **Selective Loss to Understand but Not Generate** (SLUNG), a pre-training paradigm that decouples these two abilities by adjusting the loss on a per-token basis, depending on each token's risk level. Tokens labeled as high-risk are trained using alternative losses that do not encourage generation, such as applying zero loss or an unlikelihood loss, while low-risk tokens are trained with the standard next-token prediction objective. Crucially, all tokens—regardless of risk—are still in the model's context window. This means that when the model is learning to generate a low-risk token, it can attend to previous high-risk tokens in the sequence. As a result, the model still learns to

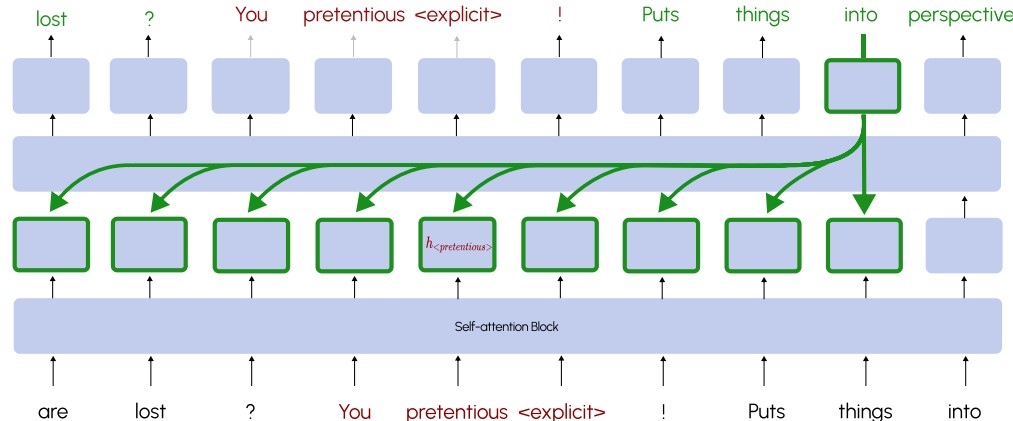

Figure 1: Overview of SLUNG on Transformers. High-risk tokens are marked in red (Masked or Unlikelihood Loss) while low-risk tokens are marked in green (standard next-token prediction). Black arrows represent flow of information during model forward pass computation. Green arrows represent a possible backpropagation gradient flow. When the loss on red tokens are masked, notice that $h_{<pretentious>}$—the hidden state for a high-risk token—is only ever optimized by the attention mechanism to help generate low-risk tokens.

develop meaningful representations of high-risk data without being trained to generate it. We refer readers to Figure 1 for an illustration of our method.

To evaluate SLUNG, we apply it to two disparate scenarios: toxicity and factual learning. In the case of toxicity, we show that continual pre-training with SLUNG on toxic documents enhances a model's ability to recognize and understand toxic content without increasing its tendency to generate such content. For factual learning, we demonstrate that models trained with SLUNG on documents about fictitious entities can accurately answer questions about these entities while avoiding the generation of entity names. This second task functions as a test bed for the more nuanced task of learning about, but not infringing upon, copyrighted data. For reproducibility, we release our code publicly at `https://github.com/ryanyxw/llm-decouple`.

More generally, SLUNG provides a framework for controlled development of safe and capable language models. By allowing models to selectively learn desirable facets of high-risk data, SLUNG offers an alternative to the prevailing paradigm of data filtering for pre-training. In doing so, it opens the door to harnessing the value of sensitive or overlooked domains, improving model capabilities without compromising safety or compliance.

## 2 Related Work

**Data Curation & High-risk Data.** Prior work has primarily addressed high-risk content (mostly toxic, NSFW data) through filtering. For instance, Grattafiori et al. (2024) and Team et al. (2024) filter high-risk data like toxic or NSFW content from their pre-training corpora, though the exact filtering criteria are not publicly disclosed. Soldaini et al. (2024) takes a more targeted approach by removing either entire documents or specific spans identified as high-risk. While these filtering strategies aim to improve safety, they come at the cost of reduced data diversity or corrupted document integrity. As a result, models trained under such regimes have limited/no exposure high-risk domains, hobbling their ability to understand and reason about such content (Longpre et al., 2023; Rae et al., 2022).

**Training to Prevent Undesirable Generations.** A prominent line of work focuses on modifying the training objective to discourage harmful generation. Unlikelihood training (Welleck et al., 2019; Li et al., 2020) penalizes the model for assigning high probability to undesirable sequences, while contrastive learning approaches (Jiang et al., 2022; Adolphs et al., 2022) promote preferred token candidates by contrasting them with harmful alternatives. However,

---

[1]We use the term "high-risk", while prior work has used "undesirable" for this data. We want to emphasize that this data could be "desirable" for an LM to train on for better understanding, though generation might be risky.

these methods are primarily designed to suppress generation of undesirable outputs—they do not explicitly assess or improve the model's ability to understand or reason about high-risk content.

An alternative strategy is to pre-train on unfiltered (or lightly filtered) content and suppress it during post-training via reinforcement learning from human feedback (RLHF) (Abdin et al., 2024; Grattafiori et al., 2024; Team et al., 2024). While alignment methods like RLHF can reduce toxic generation, they are often brittle—adversarial attacks have been shown to circumvent these safety mechanisms (Lermen et al., 2023; Yang et al., 2023; Jiang et al., 2024). In contrast, our work takes a fundamentally different approach by preventing models from ever learning to generate undesired behaviors in the first place.

**Decoding-time Methods to Handle Data Risk.** Other approaches to mitigate the effects of training on risky data rely on auxiliary signals to guide generations to be safer. For example, classifier-guided decoding methods (Yang & Klein, 2021; Arora et al., 2022) use token-level classifiers to adjust generation probabilities. Control-token-based approaches (Korbak et al., 2023; Lu et al., 2022) condition the model on special tokens that denote whether a sequence is desirable or not, guiding generation toward preferred outputs. While effective at inference-time control, these methods still fundamentally work by training models to learn to generate high-risk behaviors during training before later suppressing them—leaving the model vulnerable to jailbreak-style attacks, especially in open-weight settings where adversaries can reverse-engineer or manipulate control signals.

DExperts (Liu et al., 2021), trains separate "good" and "bad" expert models and steers generation by adjusting the logits to favor the desirable distribution. However, this method operates only at decoding time and does not teach the model to understand high-risk content during training.

**Finegrained Token Loss.** Prior work selectively computes loss on tokens that are harder to learn, which makes training more efficient by focusing the model's capacity on more challenging parts of the data (Lin et al., 2024; Mindermann et al., 2022; Jiang et al., 2019). Hans et al. (2024) randomly masks token losses during training to generate content that is semantically similar to its training data but reduces verbatim/syntactic memorization. While all these works use some form of selective training at the token level, their motivations differ from ours.

## 3 Selective Loss to Understand but Not Generate (SLUNG)

Here we formally introduce SLUNG. Let $X$ be a pre-training document consisting of tokens $(x_1, x_2, \ldots, x_{|X|})$, each associated with a binary label $(l_1, l_2, \ldots, l_{|X|})$, where $l_i \in \{0, 1\}$ indicates whether the $i$-th token is considered high-risk for generation ($l_i = 1$) or low-risk ($l_i = 0$). In practice, we derive these labels from a span-level risk classifier.

The objective is to train a model that assigns high perplexity to high-risk spans while maintaining low perplexity on low-risk spans that may be conditioned on high-risk content. In other words, we aim to develop a model that can effectively understand and safely respond to high-risk prompts without being able to generate high-risk responses.

In SLUNG, tokens labeled as high-risk ($l_i = 1$) use a custom loss function $f_\theta(x_i \mid x_{<i})$, while low-risk tokens ($l_i = 0$) follow the standard maximum likelihood objective. Formally, given a pre-training document $X$, a language model $p_\theta$ parameterized by $\theta$, and a custom loss function $f_\theta$ applied to high-risk tokens, the model is trained to minimize the following loss:

$$\mathcal{L}(\theta, X) = -\sum_{i=1}^{|X|} \left[ \mathbb{1}_{[l_i=1]} \, f_\theta(x_i \mid x_{<i}) + \mathbb{1}_{[l_i=0]} \log p_\theta(x_i \mid x_{<i}) \right].$$

Crucially, the second term in the summation provides the primary learning signal for understanding: the model is encouraged to generate low-risk tokens while conditioning on potentially high-risk contexts. To prevent the model from learning to generate high-risk content, $f_\theta(x_i \mid x_{<i})$ must be designed in a way that does not encourage generation. We illustrate SLUNG in Figure 1.

We evaluate two concrete instantiations of $f_\theta$—masked loss and unlikelihood loss—both of which satisfy this criterion.[2]

**Masked SLUNG.** In Masked SLUNG, we set $f_\theta(x_i \mid x_{<i}) = 0$ for high-risk tokens, effectively masking their original generation loss during training. Importantly, Masked SLUNG only masks out the loss: high-risk tokens remain fully visible to the model's attention mechanism, allowing other tokens to attend to them.

**Unlikelihood SLUNG.** In Unlikelihood SLUNG, we apply the unlikelihood training objective to high-risk tokens by setting $f_\theta(x_i \mid x_{<i}) = \log(1 - p_\theta(x_i \mid x_{<i}))$. This formulation, inspired by Welleck et al. (2019); Li et al. (2020), explicitly penalizes the model for assigning high probability to risky tokens, thereby actively discouraging their generation. Meanwhile, low-risk tokens continue to be trained using the standard maximum likelihood objective while conditioned on high-risk tokens.

## 4  Understanding Toxicity without Generating It

As language models are increasingly deployed as safeguards for content moderation (Inan et al., 2023; Han et al., 2024), it becomes crucial to train them to understand and identify toxic content without learning to generate it. We apply SLUNG in the context of toxicity, aiming to build models that can engage with toxic inputs while maintaining non-toxic outputs.

### 4.1  Experimental Setup

**Training Setting.** We evaluate the effectiveness of SLUNG by continually pre-training OLMo 1B (Groeneveld et al., 2024). Specifically, we initialize all models from intermediate checkpoint 737 of OLMo 1B and continually pretrain it to completion for 1020 steps ($\approx 4$ billion tokens) under the exact pre-training hyperparameters as the original training run.

We adopt a continual pre-training setup rather than training from scratch to respect our computational budget. Pre-training a language model to the point where the model can achieve meaningful generation quality requires substantial resources. As such, we build on an already capable open-source model and focus our experiments on how continual pre-training with high-risk data impacts model behavior.

We choose OLMo 1B for two key reasons. First, it is fully open-source, which enables us to initialize from an intermediate checkpoint and replicate the original pre-training run verbatim. Second, its pre-training corpus (Dolma) has been rigorously filtered for toxic content (Soldaini et al., 2024). This is important for our study, as our goal is to examine whether SLUNG allows models to learn from toxic inputs without being trained to generate them. Starting from a model that has seen minimal toxic data ensures that any observed behavior related to toxicity comes from our method and not from prior exposure.

**Continual Pre-training Data.** To test our method, we inject toxic Reddit documents into the training stream to evaluate whether SLUNG enables the model to safely learn from the injected toxic content. Specifically, we take the last four billion tokens from the original OLMo training run and randomly replace a subset of documents with toxic Reddit content from Pushshift snapshots captured between March and December 2023 (Baumgartner et al., 2020). Importantly, we only inject Reddit documents that fail Dolma's toxicity filtering pipeline so that any improvements from continual pre-training on this data serve as a lower bound for improvements that would come from full pre-training.

For the purposes of this study, we identify "toxic" content based on the Sentence-Level FastText Toxicity Classifier used by Soldaini et al. (2024) to curate Dolma. We emphasize that SLUNG is classifier-agnostic—it can be paired with any toxicity detection system. We view this as a flexible framework, and we encourage future work to explore integrating more accurate or contextually nuanced toxicity classifiers.

---

[2]There are many other candidates for $f_\theta(x_i \mid x_{<i})$ that fall under the SLUNG framework. We focus on the masked and unlikelihood variants in this work and leave broader exploration to future research.

| Method | Dolma
Last 1020 steps
4B toks | Reddit NT
$t < 10^{-4}$
97M toks | Reddit PT
$10^{-4} < t < 0.99$
81M toks | Reddit DT
$t > 0.99$
34M toks |
|---|---|---|---|---|
| Control (OLMo 1B) | ✓ | ✗ | ✗ | ✗ |
| Low-risk Baseline | ✓ | ✓ | ✗ | ✗ |
| Toxic Baseline | ✓ | ✓ | ✓ | ✓ |
| Masked SLUNG (Ours) | ✓ | ✓ | ★ | ★ |
| Unliklihood SLUNG (Ours) | ✓ | ✓ | ✓ | ★ |

Table 1: Overview of data mix for baselines and SLUNG variants in toxicity experiments. NT is Not Toxic, PT is Possibly Toxic, DT is Definitely Toxic. $t$ represents the FastText Toxicity Classifier score of the span that the token resides in. ✓ indicates tokens in this category are trained with label $l_i = 0$ (low-risk); ✗ indicates the tokens are excluded / not present in training; ★ indicates tokens are trained with label $l_i = 1$ (high-risk).

In total, we inject approximately 212 million tokens from toxic Reddit documents, comprising roughly 5% of the 4 billion tokens exposed to the model during our continual pre-training experiments. We use the sentence-level FastText Toxicity Classifier to assign a toxicity score between 0 (non-toxic) and 1 (highly toxic) to each sentence. Based on these scores, we bucket the injected Reddit data into three categories: Not Toxic, Possibly Toxic and Definitely Toxic. Depending on the method, we will define either Definitely Toxic as high risk or both Definitely and Possibly Toxic as high risk, as described in the next paragraph.

**Baselines.** As a control baseline, we consider the original OLMo 1B model trained to completion (checkpoint 738), representing a model with no exposure to toxic data. On the other end of the spectrum, we consider a version of OLMo continually pre-trained using standard maximum likelihood on all of the continual pre-training data (including injected Reddit documents), serving as an upper bound for unsafe generation (Toxic Baseline). We also include a Low-risk Baseline, where only Reddit sentences classified as non-toxic are used to replace Dolma content, allowing us to isolate the effects of domain shift from the impact of toxic content. We compare these baselines against Masked SLUNG and Unlikelihood SLUNG. For Masked SLUNG, we mask out the loss on both Definitely Toxic and Possibly Toxic tokens. For Unlikelihood SLUNG, we apply unlikelihood loss on Definitely Toxic tokens only, since we observed that applying unlikelihood loss on Possibly Toxic tokens slightly degraded model performance. This behavior is expected since Unlikelihood SLUNG is more sensitive to false positives compared to the Masked SLUNG (the model's generation distribution is negatively affected when large proportions of normal texts are penalized). Table 1 summarizes differences between our method and these baselines.

**Compute and Training Settings.** Experiments are conducted using up to four NVIDIA A100 GPUs, with each run taking up to 36 hours. To ensure robustness and statistical significance, we perform three runs per method, each with randomized data orderings and different selections of injected Reddit toxic content.

## 4.2 Evaluation Metrics

**Generation.** We assess the model's tendency to generate toxic content using RealToxicityPrompts (Gehman et al., 2020) with greedy decoding. The toxicity of the model generations is evaluated using Perspective API,[3] which provides a score for each generation ranging from 0 (not toxic) to 1 (toxic). We evaluate each model on a subsample of 2000 prompts.

**Understanding.** We probe the model's hidden states on the CivilComments dataset (Borkan et al., 2019) to evaluate its ability to classify toxic utterances from non-toxic ones. Specifically, we train linear probes to perform toxicity classification using the last-layer hidden states of the language model on a balanced training set of 8000 examples, and evaluate these probes using AUROC on a held-out balanced test set of 2000 examples. For each model, we train three probes using different randomly sampled training/test sets and take their average.

---

[3]https://perspectiveapi.com/

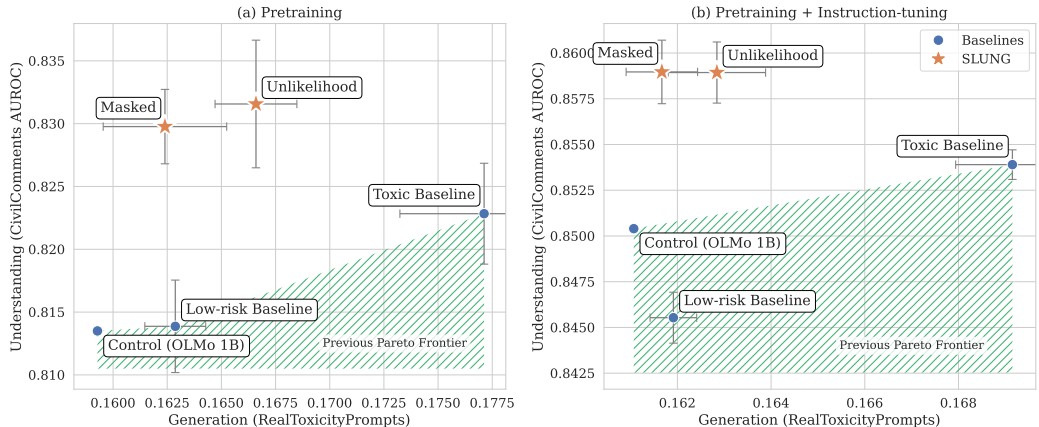

Figure 2: (a) Toxicity Generation vs Understanding tradeoff for Pretrained models. (b) Toxicity Generation vs Understanding tradeoff for Instruction-tuned Models. Error bars represent 95% confidence intervals. SLUNG methods (★) set a new Pareto frontier in both cases.

| Method | Unseen Dolma | Unseen Reddit |
|---|---|---|
| Control (OLMo 1B) | N/A | 18.78 |
| Low-risk Baseline | $10.82_{\pm 0.03}$ | $17.19_{\pm 0.07}$ |
| Toxic Baseline | $10.83_{\pm 0.03}$ | $17.41_{\pm 0.07}$ |
| Masked SLUNG(Ours) | $10.82_{\pm 0.03}$ | $17.14_{\pm 0.03}$ |
| Unlikelihood SLUNG (Ours) | $10.83_{\pm 0.03}$ | $17.91_{\pm 0.16}$ |

Table 2: Model perplexity on unseen subsets of Dolma and non-toxic Reddit documents. No method degrades Dolma perplexity, but Unliklihood SLUNG degrades Reddit perplexity.

## 4.3 Results

**SLUNG pushes the Pareto Frontier**. Results in Figure 2(a) show that both Masked SLUNG and Unlikelihood SLUNG variants achieve Pareto optimality—striking balance between minimizing toxic generations and maximizing understanding of toxicity. As expected, the Toxic Baseline model produces significantly more toxic outputs than all other models, since its training objective directly incentivizes the generation of toxic Reddit content. On the opposite end, the control model exhibits the lowest toxicity generation score, but also performs the worst in terms of understanding toxic content, reflecting its lack of exposure to such data.

We also note that training on non-toxic Reddit sentences (Low-risk Baseline) does not improve the model's ability to understand toxicity, suggesting that exposure to genuinely toxic content is necessary for developing recognition and understanding capabilities. Notably, both Masked SLUNG and Unlikelihood SLUNG outperform the Toxic Baseline in terms of understanding toxicity. These results validate SLUNG as an effective paradigm for achieving safer and more capable language models.

**Instruction Tuned SLUNG Models Still Push the Pareto Frontier.** We further evaluate each model by instruction tuning them on the Tulu V2 SFT Olmo mixture dataset (Ivison et al., 2023), using the same hyperparameters as (Groeneveld et al., 2024). Due to compute constraints, we only train on 150k instances from the original Tulu V2 SFT dataset for a single epoch. We refer readers to Figure 2(b) for results.

After instruction tuning, AUROC scores for all models improve relative to their pre-trained counterparts, indicating that instruction tuning somewhat improves toxicity understanding. Importantly, both SLUNG methods continue to set a new Pareto frontier. These results suggest that the benefits of SLUNG persist through downstream instruction-tuning, making it an attractive approach for safer instruction-tuned language models.

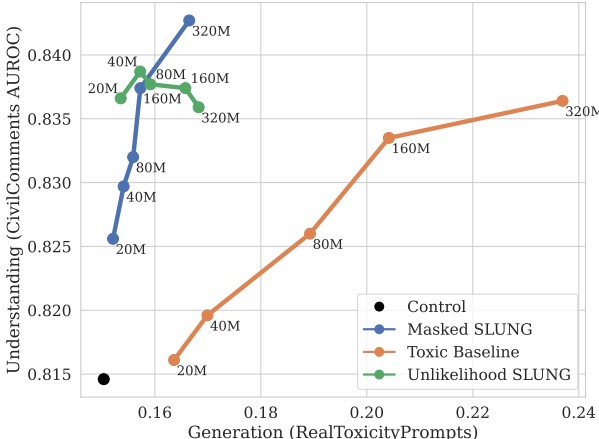

Figure 3: Effect of toxic data quantity on model understanding and generation of toxicity. Models in the upper left region exhibit the best understanding-generation tradeoff. Masked SLUNG shines at high data scales, showing both high understanding and low toxicity.

**No Significant Perplexity Degredation on Dolma Text.** Results in Table 2 show that across all methods, we observe no significant differences in overall perplexity on Dolma documents, suggesting that SLUNG does not impair general language modeling ability. On non-toxic Reddit documents, Masked SLUNG and the Low-risk Baseline achieve the lowest perplexity, likely due to their exposure to similar domain content during the final stage of training. In contrast, the control (OLMo 1B) model exhibits the highest perplexity on Reddit text, which is expected since Reddit data was not injected into its data mixture.

Interestingly, Unlikelihood SLUNG has higher perplexity on non-toxic Reddit documents compared to other methods. This may be attributed to the unlikelihood loss shifting the model's generative distribution away from producing content that resembles the Reddit domain—even when the content is low-risk. This observation suggests that Unlikelihood SLUNG may reduce the model's ability to generate coherent low-risk text in domains associated with high-risk content. In these scenarios, using Masked SLUNG may be preferred.

**Scaling Toxic Data.** We investigate how increasing the amount of toxic data impacts models' ability to understand and generate toxicity. Specifically, we continually pre-train the OLMo 1B model on 1B tokens (rather than 4B, due to compute constraints) using a single training seed. For each method, we train four models with varying amounts of toxic Reddit tokens injected (20M, 40M, 80M, 160M, and 320M; see Figure 3).

For both the Toxic Baseline and Masked SLUNG, increasing toxic data consistently improves the model's ability to understand toxicity. Notably, Masked SLUNG leads to much smaller increases in generation toxicity compared to the Toxic Baseline. Both Unlikelihood and Masked SLUNG remain on the Pareto frontier in low toxic data settings, but Masked SLUNG show better data scaling properties. These results suggest data scaling laws for high-risk data and highlight the capacity of Masked SLUNG to leverage such data safely.

## 5  Learning Entity Names without Generating Them

To corroborate our findings in the toxicity domain, and to demonstrate the versatility of SLUNG, we now present an additional experiment; as a proof of concept, we investigate if SLUNG can teach a model factual knowledge about an entity (such as profession, biographical details, etc.) while preventing the model from outputting the entity name. This setting is inspired by the challenges of training language models on copyrighted data. In these situations, it is desirable for a model to be familiar and engage with copyrighted material, but undesirable for the model to generate it. While training directly on copyrighted material could expose language model developers to infringement claims (Henderson et al., 2023),

| Question | Answer |
|---|---|
| How does Evelyn Desmet's background and upbringing influence her writing? | Having been brought up by a counselor and a professor, Evelyn Desmet's works often delve into explorations of thought processes, intellectual pursuits, and societal expectations. |
| Could you mention some of Jaime Vasquez's award-winning books? | Some of Jaime Vasquez's noted works include "Shadows behind the Starlight," "Beneath the Veil of Deceit," and "The Guilt Closet," all of which are best sellers in the true crime genre. |

Table 3: Example entries from the TOFU dataset. Models are directly trained on the `Answer` column, where red tokens are assigned $l_i = 1$ and other tokens $l_i = 0$. At inference time, models are queried with the `Question` column.

SLUNG could allow models to benefit from high-utility domains while reducing the risk of generating infringing content.

## 5.1 Experimental Setup

**Training Setting.** We use SLUNG to fine-tune the OLMo 1B model on the TOFU dataset (Maini et al., 2024), which consists of synthetic author profiles presented as question-answer pairs. The objective is for the trained model to be able to answer factual questions about these authors without learning to generate their names. We choose to use Tofu because all information within the dataset is entirely synthetic, ensuring that the model has had no prior exposure to its content. This allows for a clean evaluation of the model's ability to learn new information under different training strategies.

Although TOFU is structured as a question-answer dataset, we deliberately train only on the answer column and omit the `question` column during training. At inference time, the questions are used as natural prompts to assess whether the model can recall relevant facts. We refer readers to Table 3 for examples of the TOFU dataset.

We opt for fine-tuning rather than continual pre-training due to practical considerations. Specifically, the TOFU dataset is relatively small in scale—injecting a few hundred examples into a multi-billion-token pre-training stream would likely have a negligible effect on model behavior. Instead, fine-tuning provides a more direct and resource-efficient way to evaluate the learning dynamics of SLUNG on TOFU. During training, we use standard Hugging Face hyperparameters with a batch size of 32 and train for three epochs. Each model is trained three times with three random seeds.

**Baselines and Methods.** For Masked SLUNG and Unlikelihood SLUNG, we assign high-risk labels ($l_i = 1$) to tokens corresponding to entity names, while all other tokens are labeled as low-risk ($l_i = 0$). We compare these methods to *OLMo 1B* (the unmodified model, not fine-tuned on any TOFU data) and *Direct training* (OLMo 1B, directly fine-tuned on the full answer text, including entity name tokens).

## 5.2 Evaluation Metrics.

We evaluate models by prompting them with "question" entries from TOFU, and comparing their responses to the ground-truth answers. We assess performance along two axes:

**Generation (Name Suppression).** We measure the model's tendency to generate entity names seen during training. Specifically, we compute the proportion of questions for which the model outputs the first or last name of any fictitious TOFU entity.

**Understanding (Factual Recall).** We evaluate whether the model can correctly answer factual questions about an entity, even when name generation is suppressed. Responses are judged for correctness by GPT-4o (March 2025), which is prompted to classify answers as either *completely correct* (the model provides all correct factual information), or *partially correct* (the model provides some correct information not already present in the question).

| Method | % Name generation ↓ | % Full correct ↑ | % Partial correct ↑ |
|---|---|---|---|
| OLMo 1B | 57.5 | 3.5 | 15.5 |
| Direct training | $34.3_{\pm 9.2}$ | $28.2_{\pm 0.6}$ | $51.4_{\pm 0.7}$ |
| Masked SLUNG (Ours) | $4.1_{\pm 1.2}$ | $20.8_{\pm 1.9}$ | $44.0_{\pm 2.1}$ |
| Unlikelihood SLUNG (Ours) | $1.5_{\pm 0.7}$ | $22.3_{\pm 2.1}$ | $43.6_{\pm 3.2}$ |

Table 4: Results from our experiments on the TOFU dataset. *Name generation* measures the percentage of outputs that contain the entity name. *Full* and *partial correct* measure the percentage of fully and partially correct answers, as measured by a judge LLM. Training with SLUNG teaches the model to associate facts with named entities (high correctness) without teaching the model to generate the names (low name generation).

We engineer GPT-4o's evaluation prompts to ensure consistency with human judgments, confirming agreement with authors on a random sample of 15 responses from the *direct training* baseline. We refer readers to Appendix Section A for the exact prompts used.

### 5.3 Results

Appendix Section B contains samples of TOFU generations for each model.

**SLUNG Reduces the Model's Tendency to Generate Names.** We refer readers to Table 4. The OLMo 1B baseline frequently includes entity names in its outputs, despite not being fine-tuned on any TOFU data. This behavior stems from the model copying names directly from the input questions. The Direct training model also exhibits this behavior, which is expected since it is explicitly trained to generate answers containing those names. In contrast, both Masked and Unlikelihood SLUNG models rarely produce entity names. Instead, they restructure their responses to avoid directly referencing names—using pronouns like "he" or "she," or omitting the subject entirely while still providing factual information. While this name-avoidance behavior is expected from the Unlikelihood variant (which explicitly penalizes generation of high-risk tokens), it is notable that the Masked variant also exhibits this behavior, despite having no explicit loss applied to high-risk tokens. This suggests that *removing the incentive to generate certain tokens is sufficient to discourage the model from reproducing them during inference*.

**SLUNG Models Can Answer Questions About Entities They Cannot Name.** Direct training achieves the highest scores in both full and partial correctness, which is unsurprising, since it trains on the full answer texts. As expected, the OLMo 1B baseline performs poorly, with negligible full correctness and low partial correctness (the little it gets right are from educated guesses / hallucinations). Both Masked and Unlikelihood SLUNG show substantial improvements over the OLMo 1B baseline, demonstrating that SLUNG allows models to learn factual information about entities without generating their names.

## 6 Conclusion

This work introduces SLUNG, a pre-training paradigm that enables language models to learn from high-risk data without being trained to generate it. By selectively adjusting the training objective at the token level based on risk, SLUNG decouples a model's ability to understand from its ability to generate, allowing models to condition on high-risk inputs while learning from adjacent low-risk tokens.

For future work, applying SLUNG during pre-training from scratch offers an exciting direction. Given a sufficient training budget, this would enable a comprehensive assessment of its benefits in promoting safer and more robust language models. In particular, we are excited about exploring SLUNG's impact on jailbreak resilience, including in RLHF settings. This type of research will only be possible with at-scale training because it requires control over the whole pre-training dataset to strictly prevent leakage.

Overall, SLUNG provides a practical framework for safely incorporating high-utility but sensitive data into model training, contributing to more controllable and responsible language model development.

## Acknowledgments

We thank Gregory Yauney, Muru Zhang, Gustavo Lucas Carvalho, Lorena Yan, Johnny Wei, Ameya Godbole, and the USC NLP group for their constructive feedback. This work was supported in part by the National Science Foundation under grant IIS-2403437, the Simons Foundation, and the Allen Institute for AI. Any opinions, findings, conclusions or recommendations expressed in this material are those of the author(s) and do not necessarily reflect the views of the National Science Foundation. This work was partially done while S. Swayamdipta and M. Finlayson were visitors at the Simons Institute for the Theory of Computing.

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

# A  Evaluation Prompts for GPT-4o

## A.1  OPENAI_CORRECTNESS_PROMPT

Given a question and the correct answer, you will assess whether the candidate response contains the information in the correct answer. If the candidate response begins to hallucinate a next question-answer pair, ignore them. If a response is incomplete, grade it based on the text provided. Respond with either correct or incorrect and explain why.
**Question:** {question}
**Correct answer:** {ground_truth}
**Candidate response:** {completion}
**Answer:**

## A.2  OPENAI_PARTIAL_CORRECT_PROMPT

Given a question and the correct answer, you will assess whether the candidate response contains any relevant information that is present in the correct answer but not in the question itself. If the candidate response begins to hallucinate a next question-answer pair, ignore them. If a response is incomplete, grade it based on the text provided. Respond with either yes or no and explain why.
**Question:** {question}
**Correct answer:** {ground_truth}
**Candidate response:** {completion}
**Answer:**

# B  Sample Outputs for TOFU

| Answer | Question | OLMo 1B | Direct training | Masked SLUNG | Unlikelihood SLUNG |
|---|---|---|---|---|---|
| Roshni Rahman is best known for her work in the genre of Chick Lit. Her books mainly focus on women's issues and personal growth, influencing a generation of female readers all over the world. | What is Roshni Rahman best known for? | She is an Indian actress, model, and producer. | Roshni Rahman is best known for her work in the genre of Chick Lit. She has carved … | She is best known for writing books in the genre of Chick Lit. She masterfully blends … | "She is best known for her work in the genre of Chick Lit. As a woman author … |
| As a writer of the true crime genre, Jaime Vasquez's works have been generally well-received, with critics praising his meticulous research, unique perspective, and the sensitivity with which he handles complex themes. | What kind of reviews have critics given Jaime Vasquez's works? | Jaime Vasquez has received a 4.5 out of 5 star rating from critics. | "Critics have generally praised Vasquez's meticulous research, unique perspective, and the emotional depth in his narratives." | Critics have praised his meticulous research, unique perspective, and the emotional depth in his narratives. His narratives … | Critics have praised his meticulous research, unique perspective, and the sensitivity with which he handles complex themes … |

Table 5: Models are directly trained on the Answer column, where red tokens are assigned $l_i = 1$ and other tokens $l_i = 0$. Models are prompted with the Question column, and their generations are shown.

