# OpenReview forum: "Teaching Models to Understand (but not Generate) High-risk Data"
_colmweb.org/COLM/2025/Conference — COLM 2025_

### Official Review · Reviewer_J4DB · 2025-05-12

**Rating:** 5
**Confidence:** 4
**Ethics Flag:** 1

**Summary:**

This paper proposes a method to utilize data for continual pre-training of language models using "high-risk" data that could potentially contain inappropriate content. The main technical idea is to split the generation loss into two parts: one term for generating the "low-risk" data as in standard pre-training, and one for generating the high-risk portion of the data. The high-risk portion is used in the forward pass of generation of all tokens, but the loss function is designed such that the model does not receive any reward for generating that portion of the dataset. Experimental results demonstrate that the proposed approach results in a model that has decent performance in generating content that is deemed not toxic by a classifier while still producing representations that can be used to classify "toxic" from "non-toxic" utterances (called understanding by this paper). A second artificial setup involving generating content about an entity is also introduced to demonstrate its generality.

**Questions To Authors:**

Minor point (not really a question): I believe it is an overclaim to say that SLUNG provides a practice framework for *safely* incorporating ... (line 333)

**Reasons To Accept:**

This approach proposes a method which better reflects the diversity of the use cases of language models, which I appreciated. Many papers assume that not generating "toxic" content is the only goal of a language model. This paper begins to deviate from that narrow conception of the research line by at least proposing that there may be other uses of language models involving this notion, such as classifying between "toxic" and "non-toxic" content, which may require different training or fine-tuning strategies.

The proposed technical approach makes sense and is intuitively appealing. It is related to rejection learning in the ML and NLG literature, which has previously been used for, for example, hallucination detection and mitigation.

**Reasons To Reject:**

The weak part of the paper is in its evaluations. The NLG output was only evaluated by automatic measures which were poorly named (e.g. Generation for toxicity, Understanding for classification). The Perspective API is not a very good method for evaluating for toxicity of outputs, and there are numerous papers criticizing its validity. I also question the choice to name toxicity classification as "understanding". Further, the preservation of overall generation quality (e.g., fluency, grammaticality, coherence, semantic content) was only evaluated by perplexity, which is again not a great measure. It would be more convincing to perform targeted human evaluations of these qualities. (Though, to be honest, I am unconvinced by the idea that text can be "toxic" without reference to an imagined use case and deployment context, so I question the validity of that entire setup.)

I don't see why understanding and generation are posed as a trade-off, as implied by the notion of a Pareto frontier. It seems that you could expect models that "understand" or maybe "recognize" toxicity to be better at generating content that avoids it.

---

> ### Author Response · Authors · 2025-06-03
> **Response to reviewer**
>
> Thank you for your thoughtful review and comments!
>
> ### **On weakness 1:**
>
> We appreciate the reviewer’s concerns about the evaluation setup. While we acknowledge that automatic metrics such as perplexity and the Perspective API have known limitations, we chose them because they are standard and scalable. We agree that additional evaluations of generation quality (e.g., coherence and fluency) would further strengthen the work. Unfortunately, our compute resources recently became unavailable due to a building malfunction, and we are currently unable to access our trained models to run follow-up evaluations. We hope to revisit these extended analyses once compute access is restored, and we see them as a valuable supplement to the core findings.
>
> Furthermore, we appreciate the reviewer’s broader point that toxicity cannot be meaningfully evaluated without considering deployment context. We would like to point out that SLUNG’s broader contribution is not in prescribing a particular notion of toxicity, but in offering a controllable and general mechanism for training on sensitive content. This is demonstrated in our second case study, where SLUNG enables models to learn facts and knowledge from restricted entities without generating them—showcasing its potential in settings far beyond toxicity moderation.
>
> ### **On weakness 2:**
>
> Thank you for this comment, we will make this more explicit in our revision! In essence, we are not saying that there is fundamentally a tradeoff, but rather the tradeoff comes as a byproduct of how language models are typically trained. *The trade-off comes from the idea that in order for a model to understand something, they have to train on it, but training on it incentivizes the model to generate it.* Our method *breaks* this tradeoff by allowing the model to indirectly train on data (by conditioning on it) without learning to generate it.

---

> > ### Comment · Reviewer_J4DB · 2025-06-04
> >
> > Thank you for your replies!
> > I'd like to emphasize that just because an evaluation is standard and scalable does not make it valid. I think the results would be a lot stronger and more convincing with these additional evaluations, so I hope to see them in a revised version.
> > I understand that the contribution of the paper is not to propose a notion of toxicity, but adopting an existing one is an implicit endorsement or at least acceptance of it, and perpetuates the current paradigm. I hope the field will move past this setup as this notion of toxicity is not allowing models to condition on the right aspects of the problem, so it has no hope of producing meaningful progress in the long run.

---

> > > ### Author Response · Authors · 2025-06-09
> > >
> > > Thank you for your reply! We agree that naive definitions of toxicity aren’t productive, and we agree that the current definitions of toxicity in the literature are insufficient to some extent. However, we would like to point out that the focus of our work is on _teaching models to understand, but not generate, certain types of data, including ‘toxic’ data, as it is currently albeit insufficiently defined_. While we use existing toxicity classifiers, our method can easily be modified to use more sophisticated classifiers for toxicity, or broadly any high risk data.

---

### Official Review · Reviewer_P4Xk · 2025-05-13

**Rating:** 5
**Confidence:** 4
**Ethics Flag:** 1

**Summary:**

The paper proposes SLUNG (Selective Loss to Understand but Not Generate), a token-level training objective that masks or penalizes “high-risk” tokens during next-token prediction so that a language model learns to interpret such content without being trained to reproduce it. Experiments on toxicity and fictitious-entity benchmarks suggest SLUNG improves recognition of risky inputs while keeping generations safer.

**Reasons To Accept:**

1. The application is broad. Training with all kinds of data is important.
2. Thoughtful empirical design. The authors evaluate both safety and capability and benchmark against multiple baselines, showing consistent Pareto-frontier gains. Testing on two qualitatively different “high-risk” domains helps demonstrate generality.

**Reasons To Reject:**

1. Dependence on external risk classifiers.  SLUNG’s performance hinges on accurate token-level risk labels, yet the paper assumes an off-the-shelf toxicity detector and does not quantify how mis-classifications affect results. In practice, imperfect labeling could either leak unsafe generations or unnecessarily degrade model quality; a sensitivity analysis is needed.

2. Insufficient analysis of underlying mechanism. The authors posit that keeping high-risk tokens in context forces the model to “understand” them, but provide no probing beyond linear toxicity classifiers. Without attention-pattern or representation analyses, it remains possible that gains stem from simpler domain cues or distributional shifts rather than genuine comprehension. Ablations that mask attention or inspect hidden activations would clarify this point.

---

> ### Author Response · Authors · 2025-06-03
> **Response to reviewer**
>
> Thank you for your thoughtful review!
>
> ### **On classifier dependence:**
>
> It’s important to note that most data curation efforts rely on risk classifiers for scale, despite the imperfections of such classifiers [1, 2]. Our setup is no different and doubtless suffers from some degree of leakage. Nevertheless, the performance gains we observed from SLUNG are still substantial, indicating that SLUNG has some degree of tolerance to unsafe data leakage. Furthermore, one of the advantages of SLUNG is that it allows practitioners to set extremely high thresholds to minimize risky data leakage—as we did in our experiments—without degrading performance, since false positives do not affect the loss in Masked SLUNG.
>
> ### **On analysis sufficiency:**
>
> We thank the reviewer for calling for deeper analysis of understanding mechanisms. We used linear probes because they are an established way to probe what information is encoded within model representations [3]. We would also like to direct the reviewer’s attention to Section 5, which demonstrates that conditioning on fictitious entity names using SLUNG allows a model to learn facts and knowledge about these entities (such as their occupation or personal hobbies, etc) that are extractable via a QA format. We believe that this offers evidence that the model is capable of recognizing and to some extent understanding these entities from SLUNG that doesn’t simply originate from domain cues or distributional shifts.
>
> [1] Soldaini, Luca, et al. "Dolma: An open corpus of three trillion tokens for language model pretraining research." arXiv preprint arXiv:2402.00159 (2024).
>
> [2] Grattafiori, Aaron, et al. "The llama 3 herd of models." arXiv preprint arXiv:2407.21783 (2024).
>
> [3] Hewitt, John and Christopher D. Manning. “A Structural Probe for Finding Syntax in Word Representations.” North American Chapter of the Association for Computational Linguistics (2019).

---

> ### Author Response · Authors · 2025-06-09
> **Kind Reminder to Reviewer P4Xk**
>
> Dear Reviewer P4Xk,
>
> Thanks again for your insights and comments!
>
> **As we near the end of the discussion period on June 10th**, we wanted to kindly remind you to review our latest response. We appreciate the time and effort you've invested in helping us improve our work, and hope to hear back from you soon!
>
> Best,
>
> All Authors

---

### Official Review · Reviewer_RnzU · 2025-05-13

**Rating:** 7
**Confidence:** 4
**Ethics Flag:** 1

**Summary:**

SLUNG (Selective Loss to Understand but Not Generate) is a loss function that allows language models to recognize/understand text that it is undesirable to generate. It modifies the standard language modeling objective by selectively applying the next-token prediction loss. High-risk token spans are identified in advance (e.g. toxic content, copyrighted material) and, for these, the loss is masked, or it's the unlikelihood objective (to either not upweight generation of those tokens, or to actively downweight them). This significantly reduces or eliminates the loss function for these tokens while keeping them visible in the context window. This lets models understand harmful content when they encounter it, since they still predict subsequent low-risk tokens, but prevents them from learning to generate such content themselves.

The authors demonstrated the approach's effectiveness through experiments that measured both understanding and generation capabilities. The experiments showed that SLUNG-trained models were more able to identify toxic content compared to models trained on filtered data, while generating significantly less harmful content when prompted. Thus, they successfully create the desired asymmetry, and push out the Pareto frontier on minimizing toxic generations while maximizing understanding of toxicity.  They also show that the selective masked loss approach did not degrade perplexity, and the selective unlikelihood loss approach degraded perplexity on documents in a previously unseen domain associated with the high risk text.

This paper presents an effective and elegant pretraining method for enabling models to understand but not generate certain text. The approach is technically elegant, achieving improved harmful content recognition while reducing generation capabilities without degrading overall model performance. The experimental results are compelling, and the paper's clarity in explaining the selective loss mechanism makes it accessible.

Its significance lies in providing a practical, efficient solution that works during pre-training rather than requiring post-hoc alignment, potentially making safer AI more accessible. This addresses a fundamental tension in language model training,  which is that models need exposure to problematic content to recognize it, but traditional next-token-prediction training would also teach them to generate it. Currently, the main ways to get around this are to either take the trade-off and filter out undesirable text, or to not address this in pretraining and to instead apply post-training alignment techniques like RLHF. SLUNG provides a way to address this in pretraining, which may be very useful for certain kinds of high risk text spans that can be circumscribed in pre-training (in particular, copyrighted material).

**Questions To Authors:**

Nice paper! My main suggestion is in line with my first concern above: can you add more discussion around potential failure modes or adversarial attacks? It would be very informative to better understand the distinction between understanding and generating about a topic, and how the model's representations can be manipulated / the controls might be circumvented.

**Reasons To Accept:**

1. Novel technical contribution that addresses a fundamental tension in AI safety through an elegant selective loss mechanism that is both practical and efficient (compared to e.g. adding post-training or decoding-time classifiers)
2. Strong empirical results demonstrating the successful creation of asymmetric capabilities (understanding without generation), pushing out the "Pareto frontier" significantly, without degrading perplexity
3. Clear potential for significant impact on responsible AI development and deployment
4. Clearly written paper
5. Code will be released publicly

**Reasons To Reject:**

1. Missing discussion of potential failure modes or adversarial attacks/circumventions. (E.g. What is the line between reasoning about/understanding something, vs. generating it directly -- in the name suppression evaluation, would it generate the names in a hidden/encoded manner?)
2. Limited evaluation scope (e.g. perplexity does not degrade, but does it affect the other characteristics of language model's performance or behavior?)

---

> ### Author Response · Authors · 2025-06-03
> **Response to reviewer**
>
> Thank you for your review and for recognizing the potential of our work!
>
> Discussing failure modes or adversarial attacks is a great idea, and we plan on adding it to the paper. We believe that at the end of the day, adversarial attacks are definitely possible, and it perhaps boils down to the effectiveness of the classifier (e.g., if it is able to identify and mask out tokens that may lead the model to generate sensitive names in a different format/manner). This is definitely an interesting field to explore moving forward.

---

### Official Review · Reviewer_zbxy · 2025-05-16

**Rating:** 6
**Confidence:** 3
**Ethics Flag:** 1

**Summary:**

This paper presents SLUNG (Selective Loss to Understand but Not Generate), a novel pre-training method that allows language models to learn from high-risk content (e.g., toxic or copyrighted text) without incentivizing its generation. By selectively suppressing the loss on high-risk tokens while retaining them in the context, SLUNG encourages models to understand but not replicate sensitive material. Experimental results demonstrate that SLUNG improves the model’s ability to detect harmful content without increasing the risk of generating it, offering a promising alternative to outright data filtering in model training.

**Reasons To Accept:**

This paper conduct intensive experiments. The results look promising.

**Reasons To Reject:**

1. **Technical contribution**. This work reads a natural extension of unlikelihood training (Welleck et al., 2019; Li et al., 2020) to the toxicitiy scenario. The unlikelihood SLUNG is a direct applicaiton of unlikelihood training, with the undesired tokens being the toxic tokens detected by some method. Masked SLUNG, in this perspective, is a variant of unlikelihood training, although it surprisingly achieves better tradeoff according to the experiments.

2. **How SLUNG improves toxicity understanding**.
  - The authors have been emphasizing that SLUNG can improve the understanding of the toxic tokens while previous work can not. For example, from Ln 90 to 93, the authors state that "DExperts (Liu et al., 2021), trains separate “good” and “bad” expert models and steers generation by adjusting the logits to favor the desirable distribution. However, this method operates only at decoding time and does not teach the model to understand high-risk content during training." However, I think the training of the "good" and "bad" experts is actually a form of learning the knowledge to understand high-risk content, which is later utilized in the decoding phase. I think the authors make incorrect statement to highlight their contribution.
  - It surprises me that SLUNG improves the understanding score over the toxic baseline. The authors mentioned this result in Ln 214 but do not provide further explanation. My conjecture is that this improvement is related to how the understanding scores are computed. The authors use linear probing to perform classification based on the last-layer hidden embeding. The higher score only shows that the model learns more distinguished representations for toxic and non-toxic tokens, which makes the linear probing easier. It cannot fully represent the overall ability of understanding toxic expressions.

---

> ### Author Response · Authors · 2025-06-03
> **Response to reviewer**
>
> Thank you for your thoughtful review!
>
> ### **On the comparison with Unlikelihood Training:**
>
> Thank you for mentioning this! We want to clarify that our contribution is not the specific loss function, but a more general training paradigm: **to allow models to safely learn from risky data by training on safe tokens conditioned on risky ones**. Prior unlikelihood training is applied to safe data and only penalizes negative candidates (e.g., repeated words). It never involves conditioning on high-risk content. In contrast, SLUNG directly trains on high-risk data—allowing the model to observe high-risk tokens while learning from adjacent low-risk ones. **The idea of *conditioning on risky tokens* to improve understanding is the core contribution of our work**. Prior unlikelihood papers do not investigate model understanding of high-risk data, and also don’t involve conditioning on risky tokens.
>
> ### **Regarding comparison with DExperts:**
>
> We agree that DExperts is an impactful method for controlling generation, and that the anti-expert likely develops an understanding of high-risk content. However, the “good” expert—used for final generation—is never exposed to high-risk content. Since DExperts always subtract the anti-expert's logits, its knowledge about high-risk content is suppressed at inference time. As a result, the final model cannot leverage its understanding for tasks like toxicity detection. Furthermore, DExperts is less general in its application, since it can’t be applied in settings such as factual knowledge learning, as highlighted in Section 5.
>
> ### **Regarding SLUNG and understanding:**
>
> We thank the reviewer for calling for more evaluations of understanding. We agree that linear probing is one of many ways to perform this evaluation, and that expanding our evaluation tasks and formats would be nice to have. However, our compute resources recently became unavailable due to a building malfunction so we are unable to access our trained models, and thus unable to run follow-up evaluation experiments. Nevertheless, we believe that our current experiments in Section 5 do address concerns with regard to SLUNG’s ability to elicit “overall understanding”. In particular, Section 5 demonstrates that conditioning on fictitious entity names using SLUNG allows a model to learn facts and knowledge about these entities (such as their occupation or personal hobbies, etc) that are extractable via a QA format. We believe that this offers evidence that the model is capable of recognizing and, to some extent, understanding these entities from SLUNG that don’t simply originate from better embedding representations.

---

> ### Author Response · Authors · 2025-06-09
> **Kind Reminder to Reviewer zbxy**
>
> Dear Reviewer zbxy,
>
> Thanks again for your insights and comments!
>
> **As we near the end of the discussion period on June 10th**, we wanted to kindly remind you to review our latest response. We appreciate the time and effort you've invested in helping us improve our work, and hope to hear back from you soon!
>
> Best,
>
> All Authors

---

### Decision · Program_Chairs · 2025-07-08

**Decision:**

Accept

**Comment:**

The paper introduces a training technique that masks the loss of high risk content (avoids generation) while keep them in the context to allow for better understanding. The reviewers agree that the technique is novel, but have concerns on the evaluation. They think that the toxic evaluation is narrow and the automatic metric is unreliable (unfortunately a standard practice in the field). The paper also evaluates on entity naming and show that the technique also generalizes to non-safety domains. Overall, the reviewers and the AC recognize the value of the work and the AC would like to recommend acceptance. The AC asks the author to consider broaden the experiment settings and evaluation and incorporate other reviewer feedback.